

# A game-theoretic model for the classification of selected oil companies' price changes

Rodica-Ioana Lung* and Florin Sebastian Duma*

Center for the Study of Complexity, Babeş-Bolyai University, Cluj-Napoca, Romania
* These authors contributed equally to this work.

## ABSTRACT

One of the essential properties of a machine learning model is to be able to capture nuanced connections within data. This ability can be enhanced by considering alternative solution concepts, such as those offered by game theory. In this article, the Nash equilibrium is used as a solution concept to estimate probit parameters for the binary classification problem. A non-cooperative game is proposed in which data variables are players that attempt to maximize their marginal contribution to the log-likelihood function. A differential evolution algorithm is adapted to solve the proposed game. The new method is used to study the price changes of the Romanian oil company, OMV Petrom SA Romania, relative to the price of oil (crude and Brent) and the evolution of two other major oil companies with influence in the region. Results show that the proposed method outperforms the baseline probit and classical classification approaches in predicting price changes.

## INTRODUCTION

Machine learning methods (*Zaki & Meira, 2020*) should benefit from the use of game-theoretic solution concepts (*Maschler, Solan & Zamir, 2013*) due to their intrinsic trade-off properties. Game solutions, called equilibria, are optimal in some sense for each player and can also offer a global compromise. Different types of equilibria are defined for different game settings, for example, cooperative or non-cooperative, with perfect or imperfect information, *etc.* (*Maschler, Solan & Zamir, 2013*). They are used in many fields to model and predict the strategic behavior of agents in conflicting situations (*Tuljak-Suban, 2018*). They can also be used as alternative solutions to optimization problems in which a trade-off among several objectives is needed (*Lung, Chira & Andreica, 2014*). However, they have seldom been used as direct solutions for classification problems. Most game theoretic applications to classification are found in adversarial settings (*Dritsoula, Loiseau & Musacchio, 2017*), and some theoretical results for support vector machine models (*Couellan, 2017*). Nevertheless, if we model a classification problem as a game and find model parameters representing an equilibrium solution, we may gain new insight into the data and offer the decision-maker a novel approach.

The use of solution concepts from game theory in real-world applications is limited by several factors, among which intractability is one of the most important. This setback can

Corresponding author
Rodica-Ioana Lung,
rodica.lung@econ.ubbcluj.ro

be overcome by using computational intelligence tools such as evolutionary algorithms (*Eiben & Smith, 2015*) that can be adapted to search and approximate various types of solutions. While they may not provide exact solutions to the problem, their adaptability and scalability offer the means to find solutions that can be used in practical applications.

In this article, we propose a new game-theoretic formulation of the binary classification problem that uses the probit classification model as baseline (*Bliss, 1934*). The game assumes that variables in data are game players that search for the best probit parameters for classification. A differential evolution algorithm is adapted to compute the equilibria of this game. The approach's main goal is to estimate parameters for the probit model as the equilibrium of a game instead of as the maximum of the log-likelihood function.

As a practical application of real-world data, the proposed model is used to classify the variation of stock prices for three oil companies based on oil prices. The three oil companies are OMV Petrom SA Romania (SNP), OMV Aktiengesellschaft Austria (OMV), and Exxon Mobil USA (XOM). We used the closing price for the Brent oil and the Crude West Texas Intermediate (WTI) for almost two decades and the same period for the stocks of the three oil companies. The results provided by the proposed method are compared with probit, as well as with those provided by other classification methods.

The next section of the article presents an overview of related work. The proposed Probit variable game is described in the following section, together with the proposed method of approximating its equilibrium, probit equilibrium differential evolution. The numerical experiments section presents a brief analysis of the method's parameters and results reported on the real-world data. The article ends with conclusions and further work.

## RELATED WORK

In the field of machine learning, the binary classification problem is a central one used to build up models and explain many types of phenomena (*Srinivas, Sucharitha & Matta, 2021*; *Zaki & Meira, 2020*). Most classical binary classification approaches ultimately provide a rule for assigning label probabilities to data instances based on the information provided by some training data. In some models, such as logit or probit, probabilities are computed based on a function (*Sugiyama, 2016*), while other models, such as decision trees (*Fürnkranz, 2010*) or k-nearest neighbors (*Sam, 2010*), provide such probabilities based on the proportion of instances with a certain label in a particular region of the data space.

Game theory aims to model inherent strategic and conflicting situations and offers trade-off solutions. These solutions are called equilibria because they usually present some stability qualities, *e.g.*, against unilateral or collective deviations (*Maschler, Solan & Zamir, 2013*). There is a vast amount of literature related to the direct application of game theoretic models in very different sectors, such as medicine (*Chang et al., 2020*; *Razi et al., 2014*; *Diamant et al., 2021*), computer science (*Faugère & Tayi, 2007*; *Dasgupta & Collins, 2019*), management (*Leng & Parlar, 2005*), economy (*Pinasco, Rodríguez Cartabia & Saintier, 2018*; *Chistiakov, Andersen & Vishnevskii, 2015*), education (*Stull, 2006*), agriculture (*Gupta, Bhatt & Bhatt, 2020*), environment (*Dutta & Radner, 2006*; *Nagase & Silva, 2007*) *etc.*

Also, there is a lot of work around machine learning and game theory, mostly focused on modeling and explaining agents' behavior (*Hazra & Anjaria, 2022*; *Kiekintveld et al., 2021*; *Strumbelj & Kononenko, 2010*). However, particularly for binary classification, there are very few attempts to use equilibrium concepts as solutions for the classification problem directly. The most straightforward one is in *Couellan (2017)*, where a game based on SVMs is designed. Attempts to model the problem as a game among instances of data are made in *Suciu & Lung (2020)*. Most other applications use game-theoretic solutions to interpret or select results of other approaches and are mostly concentrated on problems that model obviously conflicting situations, such as adversarial classification (*Dritsoula, Loiseau & Musacchio, 2017*).

Regarding the energy sector, classical approaches are related to the applications of game-theoretic models to electricity markets (*Su & Huang, 2014*; *Paudel et al., 2019*; *Tushar et al., 2018*; *Lise et al., 2006*) or for natural gas (*Csercsik et al., 2019*; *Chang et al., 2021*), but not related to oil companies stocks. Interesting correlations between the oil price and the stock returns can be found in *Diaz, Molero & Perez de Gracia (2016)*, where a vector autoregressive model with several variables is estimated for the G7 economies, as well as in *Cunado & Perez de Gracia (2014)* for 15 selected European economies. Also, the effects of the oil market on the US stock market are evaluated in *Arampatzidis et al. (2021)* by estimating a structural vector autoregression model. Several other articles are studying the relationship between oil and the stock market, *e.g.*, using VAR, SVAR, or other statistical models. Other machine learning approaches to related to oil prices from different perspectives can be found in *Chen et al. (2021)*, *Guan et al. (2022)*. We could not identify articles using a game-theoretic model for this specific relation.

## THE BINARY CLASSIFICATION GAME

The binary classification problem consists in finding a rule of assigning a label, out of possible two, to some data instances, based on the information that we have about their distribution. Thus, we are given a data set $X \subset \mathbb{R}^{\mathbb{N} \times d}$, with $N$ instances $x_i \in \mathbb{R}^d$, $i = 1, \ldots, N$ and $d$ attributes/variables $X_1, \ldots, X_d$. Also, we have their corresponding labels/classes set $Y \subset \{0, 1\}^N$, such that label $y_i$ corresponds to instance $x_i$, and we want to provide a model that predicts labels $Y$ from $X$ as good as possible (*Hastie, Tibshiran & Friedman, 2016*).

The probit classification model estimates parameters that can be used to predict classification probabilities by using the normal cumulative distribution function. Parameters are usually estimated by maximizing the log-likelihood function. In this article, a new approach for estimating these parameters is proposed. The optimization problem is converted into a multi-player game among data attributes that choose a parameter that maximizes their marginal contribution to the log-likelihood function, subject to constraints. The aim is to find probit parameters that present some equilibrium properties and a good classification. Our endeavor can be empirically validated if classification results reported by using the game are better than those reported by the standard probit approach from which the approach is derived. The probit variable game and the method used to approximate its equilibrium are described in what follows.

**Probit classification** The probit classification model assumes that the probability that a label is equal to 1 can be expressed as:

$$P(Y = 1|X) = \Phi(X^T\beta) \tag{1}$$

where $\Phi$ represents the cumulative distribution function for the standard normal distribution, and $\beta \in \mathbb{R}^d$ is the model parameter, estimated by maximizing the log-likelihood function. The corresponding probability that a label is 0 is $P(Y = 0|X) = 1 - \Phi(X^T\beta)$ (*Hsiao, 1996*). The log-likelihood function $\mathscr{L}$ can be written as:

$$log\,\mathscr{L}(X, Y; \beta) = \sum_{i=1}^{N} y_i \log \Phi(x_i\beta) + (1 - y_i) \log (1 - \Phi(x_i\beta)) \tag{2}$$

The goal is to find $\beta^* = arg\,max_\beta \mathscr{L}(X, Y; \beta)$ and use it for predicting the label of an instance $x$ in the following manner:

$$y = \begin{cases} 1 & \Phi(x\beta^*) \geq 0.5 \\ 0 & \Phi(x\beta^*) \leq 0.5 \end{cases} \tag{3}$$

It is assumed that $\beta^*$ provides the best classification solution for the probit model. However, for a given problem there may be more solutions that can be used in Eq. (3) in a satisfactory manner. In this article, we explore the use of an alternative method of estimating the $\beta$ parameter for the probit model, which is based on the Nash equilibrium concept, in an attempt to offer a different and maybe more interesting solution to the classification problem.

## The probit variable game

The current approach assumes that among $\beta$ values useful for classification (Eq. (3)), we can find some that better optimize individual $\Phi(x_i\beta)$, providing a better trade-off in the maximization of their sum in $log\,\mathscr{L}$. A normal form game $\Gamma$ is designed, in which players are the variable/attributes $X_j$ in $X$ that choose their $\beta_j$ parameters to maximize their marginal contribution to the log-likelihood function, subject to satisfying conditions in Eq. (5).

Formally $\Gamma = (A, B, U)$ is defined as:

- the set of *players* $A = \{1, \ldots, d\}$: a player $j \in A$ represents an attribute $X_j$ in $X$;
- the set of *strategy profiles* $B \in \mathbb{R}^d$, an element $\beta \in B$ is $\beta = (\beta_1, \beta_2, \ldots, \beta_d)$, where $\beta_j$ is the strategy of player $j \in A$;
- the *payoff function* $U = (u_1, \ldots, u_d)$, with $u_j : B \to \mathbb{R}$:

$$u_j(\beta) = \begin{cases} log\,log\,\mathscr{L}(X, Y; \beta) - \mathscr{L}(X_{-j}, Y; \beta_{-j}) & r(\beta; X, Y) - r(\beta^*; X, Y) \geq 0, \\ log\,\mathscr{L}(X, Y; \beta) & r(\beta; X, Y) - r(\beta^*; X, Y) < 0 \end{cases} \tag{4}$$

where $X_{-j}$ represents $X$ with attribute $X_j$ removed, and $\beta_{-j}$ represents $\beta$ without $\beta_j$ and $\beta^* = arg\,max_\beta \mathscr{L}(X, Y; \beta)$. Function $r(\beta; X, Y)$ is

$$r(\beta; X, Y) = |\{i \in \{1, \ldots, N\}|x_i\beta(2y_i - 1) \geq 0\}| \tag{5}$$

where $|\cdot|$ denotes the cardinality of a set.

**The payoff function** is designed to maximize the marginal contribution of attribute $j$ to the log-likelihood function if the parameter $\beta$ evaluated improves upon the probit estimated parameter $\beta^*$ (restriction Eq. (5)), and if not, is taken as the log-likelihood function in order to be maximized. The marginal contribution is computed as the difference between $\log \mathscr{L}(X, Y; \beta)$, the value of the log-likelihood function, and $\log \mathscr{L}(X_{-j}, Y; \beta_{-j})$, the log-likelihood function computed without taking into account attribute $X_j$ (and consequently $\beta_j$) in the data. By maximizing their payoffs, either the log-likelihood will be maximized, or if a better solution can be found, it will be computed based on the marginal contribution to the log-likelihood function.

**Restrictions** $r(\beta; X, Y)$ If we consider conditions in Eq. (3), these are equivalent with having $X\beta > 0$ whenever $Y = 1$ and correspondingly $X\beta < 0$ whenever $Y = 0$. This condition can be re-written as

$$x_i\beta(2y_i - 1) \geq 0, i = 1, \ldots N \tag{6}$$

If, for a $\beta$, this condition holds, then also Eq. (3) holds, and, even if $\beta$ does not maximize the log-likelihood function, it provides a good classification of the data. But condition (6) may not hold for all $i$ even for $\beta^*$, therefore searching for a $\beta$ value such that (6) holds may be useless. However, there might exist $\beta$ values such that the number of instances for which the condition Eq. (6) holds is greater than or equal to the corresponding number computed for $\beta^*$. Among these values, we may find better classification models that are based on probit. Because we are only interested in such values, for all other situations, all players seek to maximize the log-likelihood function in Eq. (4).

**Nash equilibrium (NE)** The Nash equilibrium of a game (*Maschler, Solan & Zamir, 2013*) is a strategy profile such that no player has an incentive for unilateral deviation. This means that while all other players maintain their strategies unchanged, none of the players can improve their payoff by only changing their strategies. For game $\Gamma$, the Nash equilibrium represents a $\beta$ value such that each variable $X_j$ cannot contribute more to the log-likelihood function if all other variables maintain their choices. Finding a NE for game $\Gamma$ is not trivial. In this approach, we use a differential evolution algorithm (*Bilal et al., 2020*), adapted to approximate the Nash equilibria for game $\Gamma$ in the following manner.

## Probit equilibrium differential evolution (PrEDE)

Differential evolution (DE) is a stochastic search and optimization method that evolves a population of potential solutions to the problem (*Storn & Price, 1997*). It can be adapted to compute the Nash equilibria of a game by using the Nash ascendancy relation (*Lung & Dumitrescu, 2008*) during the selection phase of the search. We further adapt the DE to approximate the NE of game $\Gamma$, and we call this DE version Probit Equilibrium Differential Evolution (PrEDE). The outline of PrEDE is presented in Algorithm 3.

**Population and initialization** PrEDE population consists of individuals $\beta \in \mathbb{R}^d$ that are possible parameters for the probit model and strategy profiles for game $\Gamma$. Because we

---

**Algorithm 1    DE—the DE/rand/1/exp scheme to create offspring $o_i$ from parent $\beta_i$**

1:  $o_i = \beta_i$;

2:  randomly select parents $\beta_{i_1}$, $\beta_{i_2}$, $\beta_{i_3}$, where $i_1 \neq i_2 \neq i_3 \neq i$;

3:  $k = U(0, d)^1$;

4:  **for** $j = 0;\ j < d \wedge U(0,1) < CR;\ j = j + 1$ **do**

5:      $o_{ik} = \beta_{i_1 k} + F(\beta_{i_2 k} - \beta_{i_3} k)$;

6:      $k = (k+1)\% d$;

7:  **end for**

[1] $U(0, d)$ is a discrete uniform value between 0 and $d$.

---

are searching in a neighborhood of the probit parameters, the initial population is generated starting from $\beta^*$ by adding deviations following a normal distribution with standard deviation $\sigma$. Parameter $\sigma$ controls how much the initial population is spread in the search space around $\beta^*$. A very small value would lead to premature convergence to $\beta^*$, while a higher value would slow the search as the initial solutions may need a lot of improvement before satisfying restrictions in the payoff functions.

**Variation operators** A *DE/rand/1/exp* scheme, presented in Algorithm 1, is used to create offspring (*Thomsen, 2004*). With probability *CR*, some of the components of the offspring are modified based on the values of three distinct parents from the current population by adding the difference of two of them multiplied by a scaling factor *F* to the third. This is a standard DE scheme that has been proven efficient in optimization problems.

**Nash ascendancy** In order to guide the search towards the equilibrium of game $\Gamma$ the Nash ascendancy relation is used (*Lung & Dumitrescu, 2008*). Two strategy profiles of the game—here represented by the offspring *o* and parent $\beta$—are compared by counting how many players can improve their payoffs by unilaterally changing their strategy from one to the other. The strategy profile having less number of such players is considered to be better with respect to NE than the other. If the number of players that can unilaterally improve their payoffs is the same, they are considered indifferent (Algorithm 2). In the context of game $\Gamma$ an extra step is added to take into account conditions in the payoff functions (Eqs. (4) and (6)). Thus, the Nash ascendancy relation is tested only if both individuals fulfill conditions in Eq. (5), otherwise, if only one of them fulfills them, it will be considered better, and if none of them fulfills them, the one having a better probit likelihood value (Eq. 2) is considered better.

**Fitness function** While the Nash ascendancy relation is used for evolution purposes, in order to identify the best individual in the population, a specific classification-based fitness is used: the Area under the Curve (AUC) indicator (*Fawcett, 2006*) computed based on the prediction made using the cumulative normal distribution function for individuals in the population-based on training data. The AUC metric indicates the probability that a classifier will rank a positive instance higher than a negative instance. A maximum value of

**Algorithm 2** Nash ascendancy test to compare offspring o to parent $\beta$

1: $k_1 = k_2 = 0$;
2: **if** $r(o) \geq 0$ and $r(\beta) \geq 0$ **then**
3:     **for** $j = 0; j < p; j = j + 1$ **do**
4:         **if** $o_j <> \beta_j$ **then**
5:             $O' = O, \beta' = \beta$;
6:             $o'_j = \beta_j, \beta'_j = o_j$;
7:             **if** $u_j(o') > u_j(o)$ **then**
8:                 $k_1 + +$;
9:             **end if**
10:            **if** $u_j(\beta'_j) > u_j(\beta)$ **then**
11:                $k_2 + +$;
12:            **end if**
13:         **end if**
14:     **end for**
15: **else**
16:     **if** $r(o) \geq 0$ and $r(\beta) < 0$ **then**
17:         $k_2 = 1$;
18:     **else**
19:         **if** $r(o) < 0$ and $r(\beta) \geq 0$ **then**
20:             $k_1 = 1$;
21:         **else**
22:             $f_1 = L(o), f_2 = L(\beta)$;
23:             $k_1 = f_1 > f_2; k_2 = f_2 > f_1$;
24:         **end if**
25:     **end if**
26: **end if**
27: **if** $k_1 < k_2$ **then**
28:     **return** $o$ Nash ascends $\beta$ is TRUE (1);
29: **else**
30:     **if** $k_1 > k_2$ **then**
31:         **return** $o$ Nash ascends $\beta$ is FALSE ($-1$);
32:     **else**
33:         **return** $o$ is INDIFFERENT to $\beta$ (0);
34:     **end if**
35: **end if**

---

**Algorithm 3   PrEDE algorithm**

1: Generate initial population $B = \{\beta_1, \ldots, \beta_{popsize}\}$ of strategies following a normal distribution with mean probit parameter $\beta^*$ and $\sigma$;

2: Evaluate population//fitness = AUC;

3: nrgen = 0;

4: **while** (nrgen¡MaxGen) or (better $<\eta$) **do**

5:    **for** each $i = \{1, \ldots, popsize\}$ **do**

6:       create offspring $o_i$ from parent $\beta_i$ using the *DE/rand/1/exp* scheme (Alg. 1);

7:       **if** ($o_i$ *Nash ascends* (Alg. 2) parent $\beta_i$) or ($o_i$ indifferent to $\beta_i$ and fitness ($o_i$) $\geq$ fitness ($\beta_i$) **then**

8:          $o_i$ replaces parent $\beta_i$;

9:       **end if**

10:   **end for**

11:   **if** fitness of best individual better than fitness of $\beta^*$ **then**

12:      better ++

13:   **else**

14:      better = 0

15:   **end if**

16: **end while**

---

1 indicates a correct classification of the tested data. A higher value indicates a better classification from the positive label point of view.

**Termination condition** PrEDE repeats iterations until a maximum number MaxGen of iterations is reached, or if the best AUC in the population computed on the training data supersedes the AUC of the probit estimator $\beta^*$ for a successive number of $\eta$ iterations. The motivation behind stopping the search is double-fold: to reduce the computational complexity of a run and to avoid overfitting, as better AUC values on the training data may indicate both a better classification and the danger of overfitting. If a better solution is identified, and it is held for $\eta$ iterations, then it might represent a genuine improvement to probit, and it should be enough; continuing the search may indeed improve the AUC value but only for the training data, and even the probit log-likelihood function, but may not lead to an actual improvement in results.

**PrEDE parameters** PrEDE uses three types of parameters:

- parameters that are specific to any evolutionary algorithm: population size (*popsize*), and the maximum number of iterations (*MaxGen*);

- two parameters that are specific to the differential evolution algorithm: the scaling factor $F$, and crossover probability $CR$;

- two parameters specific to the classification problem: number of iterations the AUC fitness of the best individual exceeds the AUC of the probit estimator before the search is stopped $\eta$, and the standard deviation used to generate the initial population around probit parameters, $\sigma$.

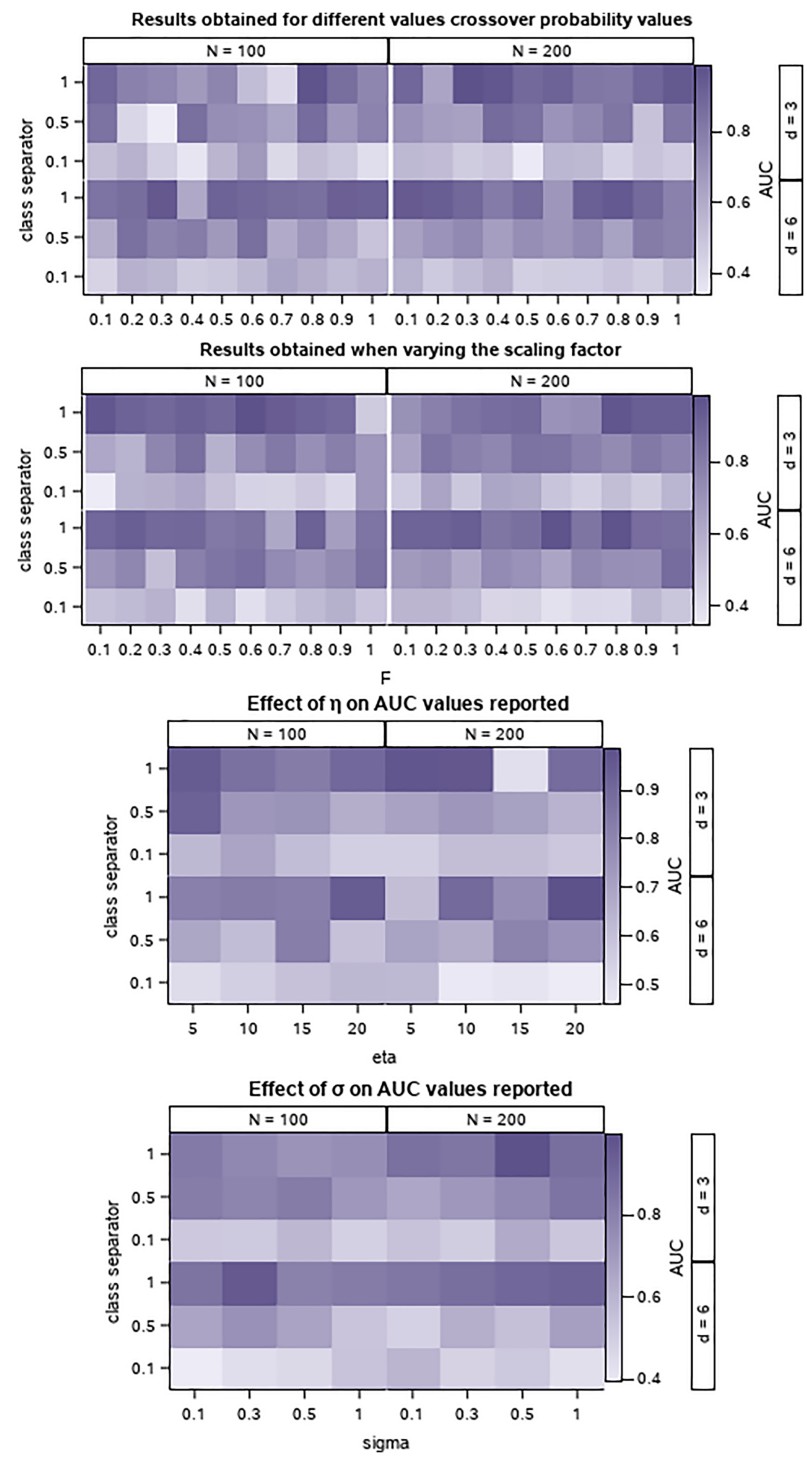

**Figure 1 Effect of parameter variation on PrEDE results for the synthetic data sets.** Colors represent average AUC values, with 1, the best, and darkest.

## NUMERICAL EXPERIMENTS

The numerical experiments section is composed of two main parts. First, results reported by PrEDE on a set o synthetic data sets are used just to illustrate the behavior of the

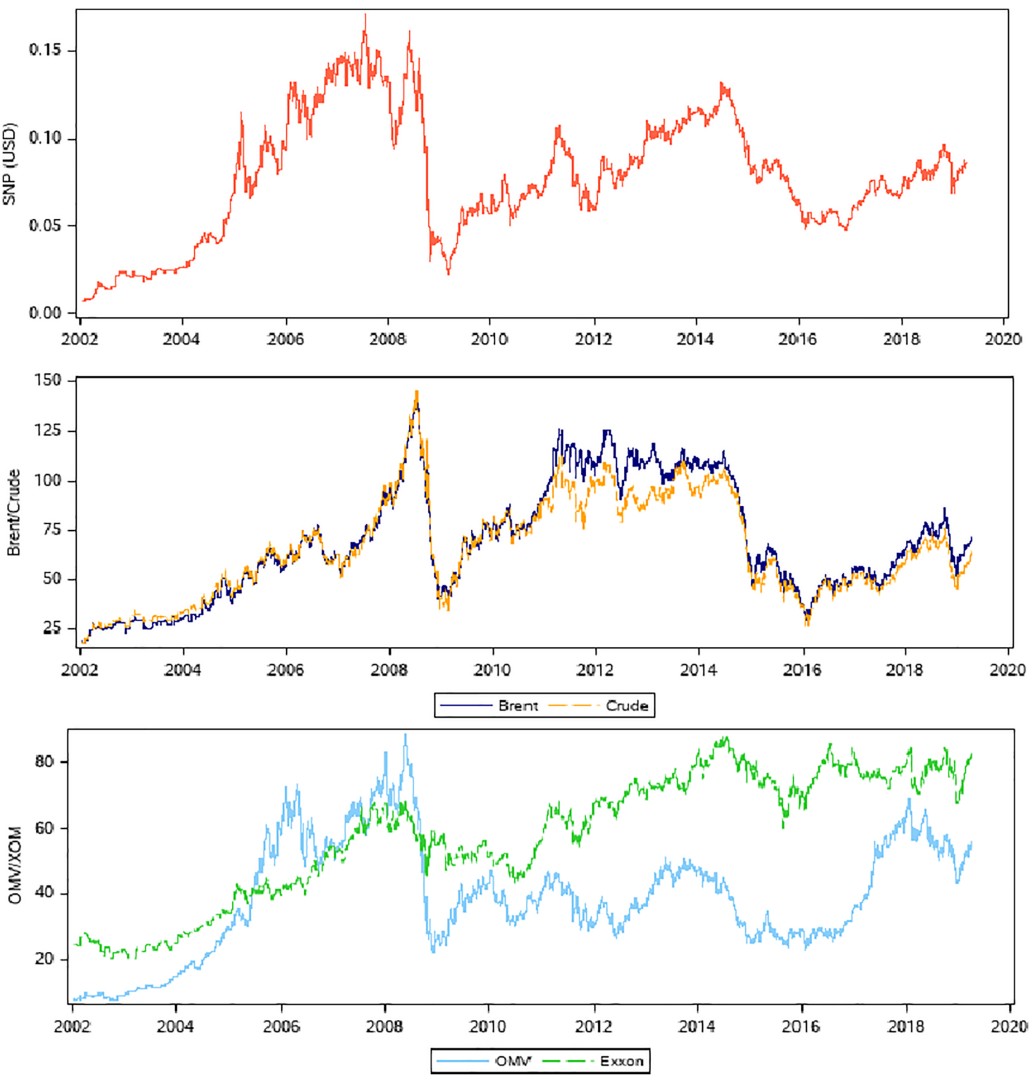

**Figure 2 Data used in the study.** Variations in SNP prices (A) are studied in relation to various combinations of oil prices Brent and crude (B) and two oil companies OMV and XOM (C).

proposed method and the effect of varying different parameters on results. In the second part, a real-world application that studies changes in the price of oil of the Romanian national oil company based on oil prices and two mainstream companies with direct or indirect influence on the region.

## Synthetic data and parameter testing

A set of synthetically generated data is used to preliminary assess the performance of PrEDE in various settings and provide an overview of the effect of parameter settings on results. The datasets are generated by using the `make_classification` function in the `scikit-learn` Python (https://scikit-learn.org/stable/modules/generated/sklearn.datasets.make_classification.html, last accessed Feb. 2022). To simulate an environment that is similar to the one created by the oil price data, we generated datasets with 100 and 200 instances, with

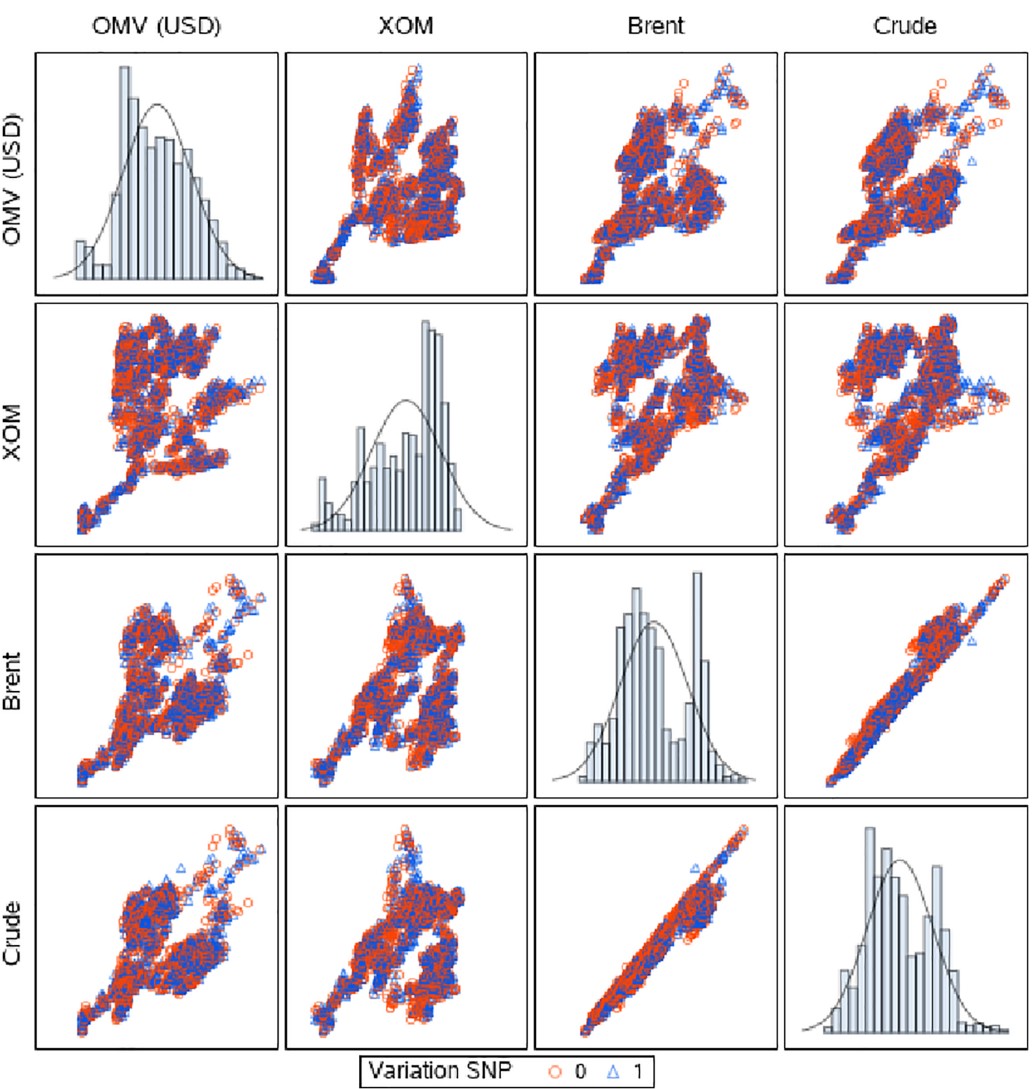

**Figure 3 Scatter matrix of data grouped by the SNP variation labels.** The two groups overlap significantly, creating a challenging classification problem.

three and six attributes, and with different degrees of separation between classes by setting the `class separator` parameter to 0.1, 0.5 and 1. A higher value creates instances with better-separated classes. As a performance indicator, the AUC (*Fawcett, 2006*) is used. AUC takes values between 0 and 1, 1 indicates a correct classification, and it can be used to compare results. A total of 10-folds cross-validation is used and the average of the 10 AUC values reported on the tested folds are presented (*Hastie, Tibshiran & Friedman, 2016*; *Stone, 1974*). Figure 1 presents the tested values and corresponding AUC values.

The differential evolution parameters, *F*, and *CR* are set to take values from 0.1 to 1 with a step of 0.1, Fig. 1. We find, as expected, that there is no 'gold' setting for these parameters. However, smaller *CR* and *F* values seem to provide a good trade-off on datasets with smaller class separator values which are more difficult to solve.

The parameter $\eta$ is used to stop the search if for a successive number of $\eta$ iterations the fitness of the best individual in the population is better than the fitness of the probit

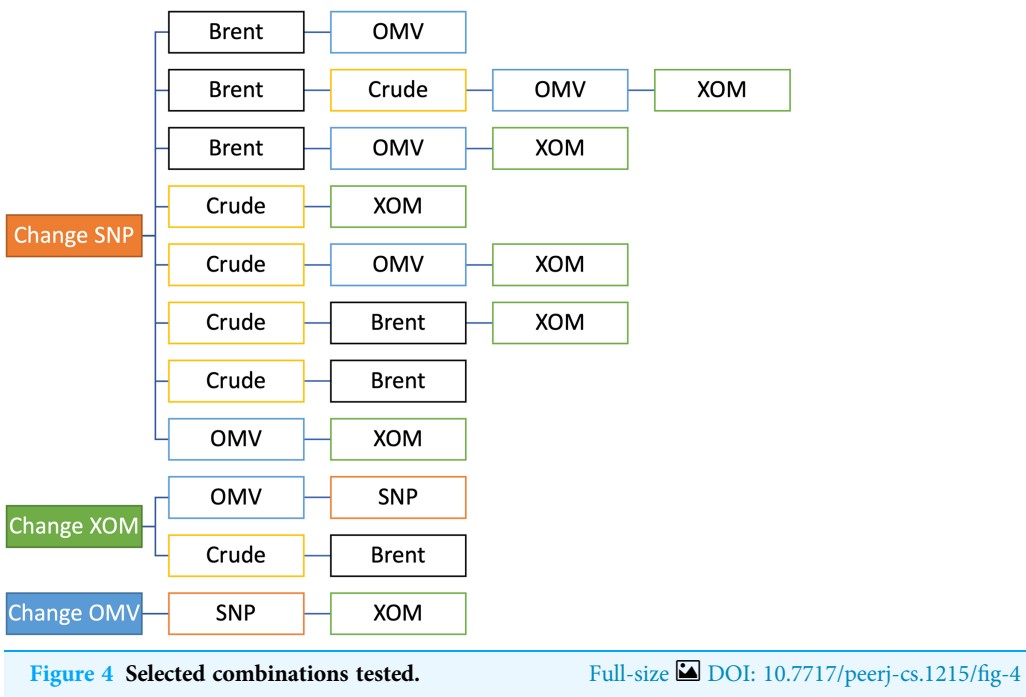

**Figure 4** Selected combinations tested. 

parameter. This ensures that the solution provided by PrEDE is indeed better than that provided by probit while avoiding overfitting (Fig. 1). The value of $\sigma$ least influences results (Fig. 1), indicating that the choice of initial population setting does not influence the search.

## Application: oil data

Data used in the study consists of the closing price for the crude oil WTI and the Brent oil collected by the authors from the Bloomberg database for almost two decades, starting from 2000 until 2019. From the same database and for the same period, the closing prices for three stocks: Exxon Mobil (XOM, Irving, TX, USA) from the US, OMV AG Austria (OMV, Vienna, Austria), and OMV Petrom SA Romania (SNP, Bucharest, Romania) were also collected. To better compare and avoid any distortions, all the data were converted into US dollars. We selected these three oil companies because they are from different geographical areas, and they are also different in size, but they have similar activities. Yet, what is more important, these companies have some close connections: OMV AG owns 51 percent of Petrom, while Petrom and Exxon have made together a joint company to prospect and exploit natural gas from the Black Sea. Other reasons for this selection were the fact that, on the one hand, the oil companies from the area of Central and Eastern Europe (in our case, OMV Petrom, Bucharest, Romania) were not researched enough yet and, on the other hand, Exxon Mobil, besides its links with OMV Petrom, is probably the most representative oil company in the world. Figure 2 illustrates the collected data used for the analysis.

    One of the issues of interest when looking at oil price data is to predict if an increase/decrease in price is expected for these three stocks. In this approach, we test the following

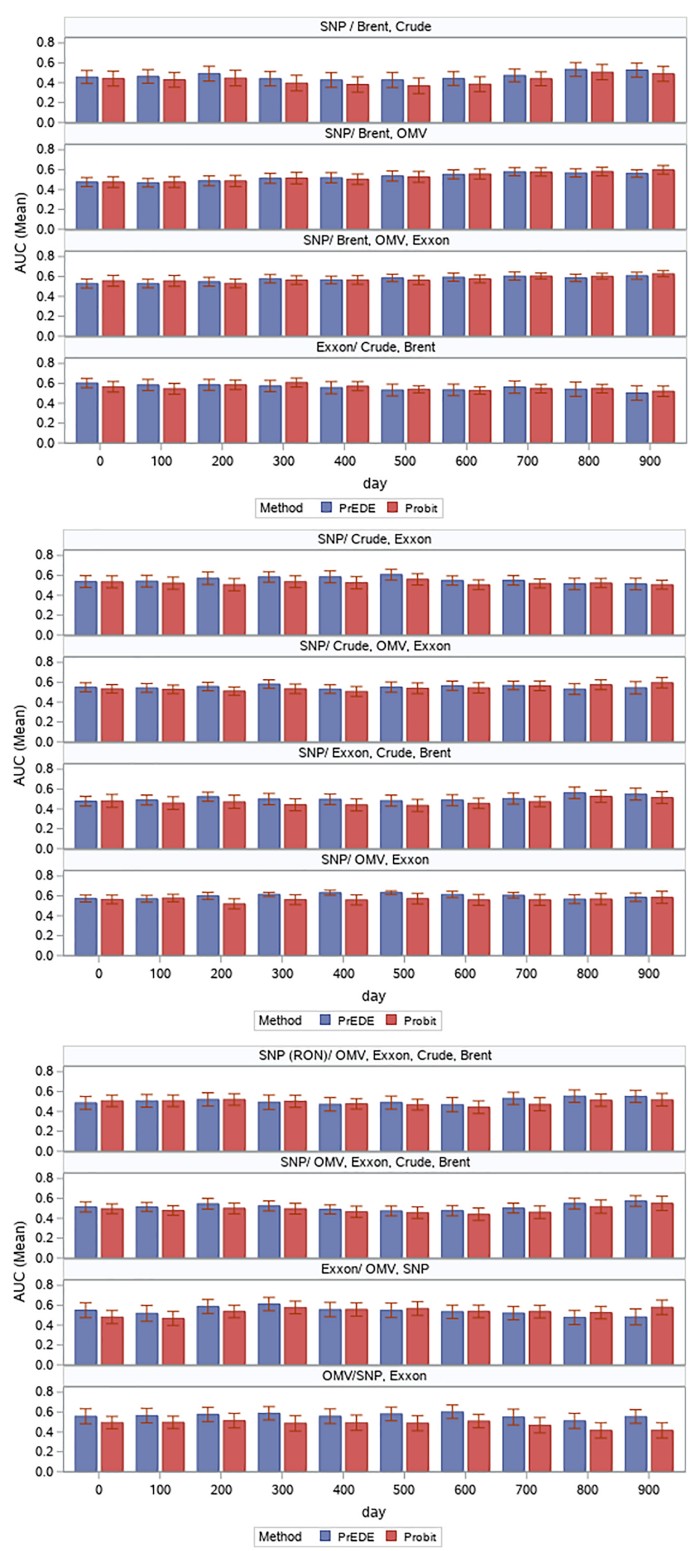

**Figure 5 Average AUC values (with error bars) reported by the two methods for each time frame.**

**Table 1 Descriptive statistics of AUC values reported by each method on the selected data.** An asterisk (*) indicates that differences in results are significant according to a paired t-test with $\alpha = 0.05$. There are no situations in which probit results are significantly better than those reported by PrEDE. An underline means that there is a result provided by another method better than the one reported here (Table 2).

| Selection | Method | Mean | Std Dev | Lower 95% | Upper 95% |
|---|---|---|---|---|---|
| SNP ↔ Brent, Crude | PrEDE | 0.47 (*) | 0.22 | 0.42 | 0.51 |
| | Probit | *0.43* | 0.23 | 0.38 | 0.47 |
| SNP ↔ Brent, OMV | PrEDE | 0.53 | 0.14 | 0.50 | 0.55 |
| | Probit | 0.53 | 0.16 | 0.50 | 0.56 |
| SNP ↔ Brent, OMV, XOM | PrEDE | 0.57 | 0.13 | 0.55 | 0.60 |
| | Probit | 0.57 | 0.13 | 0.55 | 0.60 |
| XOM ↔ Crude, Brent | PrEDE | 0.56 | 0.19 | 0.52 | 0.59 |
| | Probit | 0.56 | 0.14 | 0.53 | 0.58 |
| SNP ↔ Crude, XOM | PrEDE | 0.56 (*) | 0.17 | 0.52 | 0.59 |
| | Probit | 0.52 | 0.17 | 0.49 | 0.56 |
| SNP ↔ Crude, OMV, XOM | PrEDE | 0.55 | 0.15 | 0.52 | 0.58 |
| | Probit | 0.54 | 0.15 | 0.51 | 0.57 |
| SNP ↔ XOM, Crude, Brent | PrEDE | 0.51 (*) | 0.17 | 0.47 | 0.54 |
| | Probit | 0.47 | 0.19 | 0.43 | 0.51 |
| SNP ↔ OMV, XOM | PrEDE | 0.60 (*) | 0.10 | 0.58 | 0.62 |
| | Probit | 0.56 | 0.16 | 0.53 | 0.59 |
| SNP (RON) ↔ OMV, XOM, Crude, Brent | PrEDE | 0.51 (*) | 0.20 | 0.47 | 0.55 |
| | Probit | 0.49 | 0.18 | 0.46 | 0.53 |
| SNP ↔ OMV, XOM, Crude, Brent | PrEDE | 0.52 (*) | 0.16 | 0.49 | 0.55 |
| | Probit | 0.49 | 0.18 | 0.45 | 0.52 |
| XOM ↔ OMV, SNP | PrEDE | 0.54 | 0.22 | 0.49 | 0.58 |
| | Probit | 0.54 | 0.21 | 0.50 | 0.58 |
| OMV ↔ SNP, XOM | PrEDE | 0.56 (*) | 0.22 | 0.52 | 0.61 |
| | Probit | 0.48 | 0.22 | 0.43 | 0.52 |

model: a binary variable taking values of 0 and 1 is created for the stock prices of each oil company, with value 1 representing an increase in the stock price from the previous day and 0 a price decrease. Figure 3 presents the overlapping degree of the two obtained classes for all pairs of considered data. Probit models changes in the oil price and PrEDE, and results are compared and discussed.

Ten intervals of 50 days were selected for training, and the subsequent 10 days were used in the testing phase. This is a reasonable setting as it is common to look for the trends of the last 50 days to make predictions related to the near future. AUC values were reported for each time slot. Various combinations of data were tested in this manner. Our focus was mainly on the Romanian oil company OMV Petrom (SNP, Bucharest, Romania); thus, we tested the changes in eight different combinations with Brent and Crude oil and with the other two oil companies, combinations that we considered that are the most useful ones, as we can see from in Fig. 4. We were primarily interested in SNP because it is the least

**Table 2 Results reported by other methods for the same data (average and standard deviation of AUC values).** Only two instances of these results, highlighted in *italics*, are significantly better than those reported by PrEDE.

| Selection | Logit | kNN | RF |
|---|---|---|---|
| SNP ↔ Brent, Crude | 0.43 ± 0.23 | *0.53 ± 0.16* | *0.50 ± 0.11* |
| SNP ↔ Brent, OMV | 0.53 ± 0.16 | 0.51 ± 0.13 | 0.47 ± 0.19 |
| SNP ↔ Brent, OMV, XOM | 0.57 ± 0.13 | 0.51 ± 0.12 | 0.49 ± 0.19 |
| XOM ↔ Crude, Brent | 0.55 ± 0.14 | 0.57 ± 0.17 | 0.51 ± 0.15 |
| SNP ↔ Crude, XOM | 0.53 ± 0.17 | 0.46 ± 0.13 | 0.42 ± 0.17 |
| SNP ↔ Crude, OMV, XOM | 0.55 ± 0.15 | 0.53 ± 0.11 | 0.48 ± 0.19 |
| SNP ↔ XOM, Crude, Brent | 0.47 ± 0.19 | 0.51 ± 0.17 | 0.50 ± 0.12 |
| SNP ↔ OMV, XOM | 0.56 ± 0.16 | 0.55 ± 0.15 | 0.45 ± 0.19 |
| SNP (RON) ↔ OMV, XOM, Crude, Brent | 0.49 ± 0.19 | 0.52 ± 0.13 | 0.50 ± 0.17 |
| SNP ↔ OMV, XOM, Crude, Brent | 0.49 ± 0.18 | 0.50 ± 0.17 | 0.52 ± 0.14 |
| XOM ↔ OMV, SNP | 0.54 ± 0.21 | 0.53 ± 0.19 | 0.50 ± 0.19 |
| OMV ↔ SNP, XOM | 0.48 ± 0.22 | 0.48 ± 0.17 | 0.48 ± 0.23 |

researched of the three companies and has been less studied. In order to do this analysis, we tested SNP in combination with the oil prices and the other oil companies in pairs of two, three, or four data sets (Fig. 4). Firstly, we tested SNP stock price variation based on Brent oil, because this is the European benchmark, and together with OMV, because it is the majority shareholder of SNP, and then based on two more combinations with Brent. Secondly, we continued by testing SNP stock price variation with the crude oil, which is the US benchmark, in four ways, including Exxon, because it is a US company, then adding OMV, and then Brent. Thirdly, we tested SNP stock price variation based on the stock prices of the other two oil companies, OMV and XOM. In the end, mainly for confirmation purposes, we tested XOM and OMV stock price variation, in relation with each-others, with the two oil benchmarks, and with SNP.

Results are presented as error bars of AUC values in Fig. 5 and corresponding numerical values in Table 1. While values are comparable, we find the PrEDE reports better average AUC values than probit in eight out of the 12 scenarios tested. Figure 5 illustrate AUC values for each set of days tested. A paired t-test performed overall AUC values for each data set, and each starting date shows a significant difference between AUC values reported by PrEDE and probit ($t = 5.86$, and $p < 0.0001$). For seven out of the 12 combinations tested, the differences between results are significant, while there is no situation in which probit results can be considered better. A detailed representation of differences is illustrated in Fig. 5. Considering the high overlapping degree of the data (Fig. 3), it is to be expected that average AUC values are around 0.5; however, the fact that PrEDE has been able to improve results reported by probit indicates the potential of exploring such an approach. Results reported by other standard classification methods are presented in Table 2. Logit (*Seabold & Perktold, 2010*), k-nearest neighbor (kNN), and a random forest (RF) (*Pedregosa et al., 2011*) were used on the same data, and the mean and standard

deviation of AUC values are reported. Only in one instance (SNP ↔ Brent, Crude) kNN and RF results were significantly better than those reported by PrEDE.

## CONCLUSIONS

Game theory and machine learning are naturally considered interconnected, with many data models attempting to use game theory concepts to explain results or agents' behavior. However, there are various other ways in which game theory can be involved in machine learning. The direct use of equilibria as solution concepts when estimating model parameters has not yet been explored, despite the advantages provided by their intrinsic trade-off capabilities.

An example of such use of equilibrium is presented in this article. A non-cooperative game is designed in such a manner that the game strategies, and hence its equilibrium, are probit parameters. Players of the game aim to improve upon probit parameters by maximizing a payoff based on their attribute's marginal contribution to the log-likelihood function. The solution of the game is represented by parameters such that none of the players can unilaterally improve its marginal contribution to the log-likelihood function. An equilibrium of this game is approximated by a stochastic search method based on a differential evolution algorithm adapted to solve this game.

Thus, the goal of our endeavor is to show that there are other solution concepts that can be explored within a classical classification framework. While we assume that maximum likelihood methods may provide the best possible classification of data based on a particular model, such as probit, there may be some other parameters, endowed with different trade-off properties, that, under the same model, offer a better classification for some data.

The limitation of such an approach is, for the moment, at a theoretical level, as an in-depth analysis of such alternatives to probit is required to generalize results. However, the flexibility provided by a search heuristic such as the differential evolution, which has been adapted in the context of the probit variable game, consists in its flexibility: it can be adapted to other game settings and other data to provide a different—equilibrated—insight into its structure.

The new approach is tested on a set of real oil data collected between 2002 and 2020 to study the influence of the oil price and the prices of two major companies on the price changes of the Romanian national oil company. We find data to be highly overlapping and consider it a challenge from a classification point of view. Nevertheless, PrEDE improves upon probit on these data, indicating that the game-theoretic approach has the potential to uncover better relationships within it. A future research direction could be to extend the use of this new method to investigate the price changes of the Romanian OMV Petrom SA shares relative also to its peers from Central and Eastern Europe like PKN Orlen from Poland, MOL from Hungary, Unipetrol from Czech Republic and to some other regional companies from the same field. Other similar analyses may also be envisaged.

### Funding
This work was supported by a grant from the Romanian Ministry of Education and Research, CNCS—UEFISCDI, project number PN-III-P4-ID-PCE-2020-2360, within PNCDI III. The funders had no role in study design, data collection and analysis, decision to publish, or preparation of the manuscript.

### Grant Disclosures
The following grant information was disclosed by the authors:
Romanian Ministry of Education and Research, CNCS—UEFISCDI: PN-III-P4-ID-PCE-2020-2360, within PNCDI III.

### Competing Interests
The authors declare that they have no competing interests.

### Author Contributions
- Rodica-Ioana Lung conceived and designed the experiments, performed the experiments, analyzed the data, performed the computation work, prepared figures and/or tables, authored or reviewed drafts of the article, and approved the final draft.
- Florin Sebastian Duma conceived and designed the experiments, performed the experiments, analyzed the data, prepared figures and/or tables, authored or reviewed drafts of the article, and approved the final draft.

### Data Availability
Raw data and code are available in the Supplemental Files.

### Supplemental Information
Supplemental information for this article can be found online at http://dx.doi.org/10.7717/peerj-cs.1215#supplemental-information.

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
