# Peer review of "A game-theoretic model for the classification of selected oil companies’ price changes"

_PeerJ Computer Science, doi:10.7717/peerj-cs.1215_

## Round 0.1 · original submission · Major Revisions

The reviewers have suggested revising the paper. Kindly go through the reviewers' comments and address the suggestions!

Reviewer 1 ·

Basic reporting

The authors have presented a game theoretic model for binary classification of price changes in oil companies. For this purpose, a probit classification problem is studied and the parameters in this model are estimated that provides the state of a Nash equilibrium. A stochastic search algorithm, known as the differential evolution, is used to solve for the Nash equilibrium. Finally, numerical experiments using price changes of the Romanian oil company and 2 other oil companies are used to validate the framework.

The paper is technically well-written (except for a few typos like on Page 2, before Eq (2) "ca" should be "can", and space is missing before and after "and" in Eq. (3). )

It is not clear from the introduction about the deficiencies in existing algorithms for game-theoretic classification models that the authors are trying to address. References on several game theoretic approaches are missing (including differential games). Moreover, the introduction starts with Machine Learning approaches but then the discussion shifts to game theoretic approaches. Subsequent discussion on different machine learning approaches for classification are not discussed and made clear, which is misleading. It will be good to have some discussion to highlight the major approach presented in the paper.

The results presented, with both synthetic and real data, are thorough and correct, and the raw codes have been made available as supplementary material.

Experimental design

It is not clear how the authors have improved on the existing literature with their presented method. The classification game and Nash equilibrium concepts are quite classical and well-established. It is important to clearly state the problem at hand, the existing methods available to solve the problem and its deficiencies, and the novelty of the author's contribution.

Validity of the findings

Though the results presented are corrected and can be replicated using the supplementary materials, as per the discussion in the previous sections, the novelty of the work is not clearly mentioned.

Additional comments

No comments.

Reviewer 2 ·

Basic reporting

1. Please explain the abstract of the paper in a better way.
2. The authors didn't write the introduction part appropriately.
3. The aim, method, and problem are not described clearly and scientifically. The equations are not described. The results are not described. Most of the abbreviations are not given. The proposed method is not described.
4. Citation of reference papers is not in order of lines 18, 20, and 22.
5. When citing related work citations that are not done correctly, please rely on recent references.
6. The manuscript needs to be corrected for modifications to the grammar as well as checked.
7. I think the distance between the monitor screen to the respondent's eye should be less than 100 cm.
8. In some parts of the paper, the clarity and editorial quality of the paper are compromised. As a consequence, such parts result in being quite challenging to read. Therefore, I would suggest carefully improving the writing prose to make this paper easier to read.
Bullet 1) is critical, and the authors must highlight and compare with similar papers the proposal.
9. The conclusion must be improved and avoid redundancy.

Experimental design

No comment

Validity of the findings

No comment

Additional comments

No comment

---

## Round 0.2 · Minor Revisions

Kindly incorporate the minor revisions suggested by Reviewer 2.

Reviewer 1 ·

Basic reporting

The authors have very well taken into account the previous comments and concerns, and substantially improved the manuscript. I now recommend it for publication.

Experimental design

The experimental design is adequate and well presented.

Validity of the findings

The results are correct and support the framework present in the paper.

Reviewer 2 ·

Basic reporting

In this paper, the authors present A game-theoretic model for the classiûcation of selected oil companies' price changes. The paper presents he Nash equilibrium is used as a solution concept to estimate probit parameters for the binary classiûcation problem.

Some suggestions to improve the paper are as below:
1.The abstract can be rewritten to be more meaningful. The authors should add more details about their final results in the abstract. Abstract should clarify what is exactly proposed (the technical contribution) and how the proposed approach is validated.
2. Also in the Introduction, Sections must be mentioned in the introduction.
3. What are the main limitations of the work?
4. Proposed methods should be compared with the state-of-the-art existing techniques.
5. Limitations and Highlights of the proposed methods must be addressed properly.
6. Overall, the paper is very well written, and the information is interesting from the field under study. However, some important points need to be take:
a) Bullet 2) is critical, and the authors must highlight and compare with similar papers of the proposal.
7. The conclusion must be improved and avoid redundancy. The conclusion provide some insights into the outcome of the paper. However, I feel that we need further elaboration and critical evaluation within the conclusions.
8. For easy follow-up: A Table should be provided to compare your results with the literature.
9. It is preferable to mention future works in thies paper?
10. The quality of the figures needs to be improved.
11. References: Critical: update it.
Finally, paper needs mainor improvements

Experimental design

no comment

Validity of the findings

no comment

Additional comments

no comment

---

## Round 0.3 · accepted · Accept

I am pleased to say that the authors have addressed the reviewers' main comments and the manuscript is now ready for publication.

Reviewer 2 ·

Basic reporting

no comment

Experimental design

no comment

Validity of the findings

no comment

Additional comments

In this paper, the authors present A game-theoretic model for the classification of selected oil companies' price changes. The paper presents the Nash equilibrium is used as a solution concept to estimate probit parameters for the binary classiûcation problem. A non-cooperative game is proposed in which data variables are players that attempt to maximize their marginal contribution to the log-likelihood function.

1. The Abstract section should summarize in a very concise way the main achievement of the paper. Describe vague ideas about the main results. The authors are instigated to highlight what is new in their studies.
2. The language needs minor review to improve readability.
3. Also in the Introduction, it is important to enhance your motivation. Where the current surveys fail.
4. Authors must develop the framework/architecture of the proposed methods
5. Proposed methods should be compared with the state-of-the-art existing techniques
6. In case to be allowed, it would be interesting authors create a list of abbreviatures.
7. What are the main limitations of the work?
8. Overall, the paper is very well written, and the information is interesting from the field under study. However, some important points need to be take:
a) Bullet 1) is critical, and the authors must highlight and compare with similar papers of the proposal.
9. The quality of the figures needs to be improved.
10. Some of the definitions included in the article should be improved for a better understanding by the reader since some are very basic.
11. Experimental results are not convincing, so authors must give more results to justify their proposal.
12. References: Critical: update it.
Finally, paper needs major improvements

---

## Author Rebuttal · Round 0.3

Dear Editor and Reviewers,

We would like to thank you for considering our paper for publication and for the suggestions that helped us to improve our presentation. We have addressed all the issues raised by Reviewer 2 and included them in the paper. Please find below a more detailed answer.

With respect,

Rodica Lung,

On behalf of the authors
* * *
# Reviewer 1 (Anonymous)

## Basic reporting

The authors have very well taken into account the previous comments and concerns, and substantially improved the manuscript. I now recommend it for publication.

## Experimental design

The experimental design is adequate and well presented.

## Validity of the findings

The results are correct and support the framework present in the paper.

Thank you for your remarks.

# Reviewer 2 (Anonymous)

## Basic reporting

In this paper, the authors present A game-theoretic model for the classiûcation of selected oil companies' price changes. The paper presents he Nash equilibrium is used as a solution concept to estimate probit parameters for the binary classiûcation problem.

Some suggestions to improve the paper are as below:

1.The abstract can be rewritten to be more meaningful. The authors should add more details about their final results in the abstract. Abstract should clarify what is exactly proposed (the technical contribution) and how the proposed approach is validated.

> The abstract has been modified. It presents the main approach of the paper as well as a note related to the results.

2. Also in the Introduction, Sections must be mentioned in the introduction.

> We added a paragraph describing the structure of the paper in the Introduction.

3. What are the main limitations of the work?

> The limitations are related to the theoretical aspects using equilibria parameters, as we do not have an in-depth understanding of their behavior other than that offered by the results of the numerical experiments which cannot be generalized. We pointed that out in the paper.

4. Proposed methods should be compared with the state-of-the-art existing techniques.

> Numerical results provided by PrEDE and Probit are complemented with those reported by Logit, KNN and RandomForest classifiers in Table 2, with comments related to the significance of differences.

5. Limitations and Highlights of the proposed methods must be addressed properly.

> We have addressed these issues within the Conclusions.

6. Overall, the paper is very well written, and the information is interesting from the field under study. However, some important points need to be take:
a) Bullet 2) is critical, and the authors must highlight and compare with similar papers of the proposal.

> We have addressed it.

7. The conclusion must be improved and avoid redundancy. The conclusion provide some insights into the outcome of the paper. However, I feel that we need further elaboration and critical evaluation within the conclusions.

> We have modified the Conclusions.

8. For easy follow-up: A Table should be provided to compare your results with the literature.

> Table 2 presents results reported by other methods.

9. It is preferable to mention future works in thies paper?

> We have mentioned possible future research directions in the Conclusions.

10. The quality of the figures needs to be improved.

> We have made all the efforts to abide by the journal's rules with regards to the quality of the figures.

11. References: Critical: update it.

We have added some new references related to oil prices.

Finally, paper needs mainor improvements

# Experimental design

no comment

# Validity of the findings

no comment

# Additional comments

no comment

Thank you for your helpful comments.